# Hierarchical Analysis Process for Belief Management in Internet of Drones

**DOI:** 10.3390/s22166146

**Published:** 2022-08-17

**Authors:** Hana Gharrad, Nafaâ Jabeur, Ansar Ul-Haque Yasar

**Affiliations:** 1Transportation Research Institute (IMOB), Hasselt University, 3500 Hasselt, Belgium; 2Computer Sciences Department, German University of Technology in Oman (GUtech), Athaibah, Muscat 130, Oman

**Keywords:** collaborative awareness, uncertainty, belief fusion, drone collaboration, belief classification

## Abstract

Group awareness is playing a major role in the efficiency of mission planning and decision-making processes, particularly those involving spatially distributed collaborative entities. The performance of this concept has remarkably increased with the advent of the Internet of Things (IoT). Indeed, a myriad of innovative devices are being extensively deployed to collaboratively recognize and track events, objects, and activities of interest. A wide range of IoT-based approaches have focused on representing and managing shared information through formal operators for group awareness. However, despite their proven results, these approaches are still refrained by the inaccuracy of information being shared between the collaborating distributed entities. In order to address this issue, we propose in this paper a new belief-management-based model for a collaborative Internet of Drones (IoD). The proposed model allows drones to decide the most appropriate operators to apply in order to manage the uncertainty of perceived or received information in different situations. This model uses Hierarchical Analysis Process (AHP) with Subjective Logic (SL) to represent and combine opinions of different sources. We focus on purely collaborative drone networks where the group awareness will also be provided as service to collaborating entities.

## 1. Introduction

Smart cities applications are built open distributed sensors and equipment like sensors embedded in infrastructure, vehicles, UAVs, etc. These devices could collect useful data which can be integrated and analyzed to infer meaningful information and improve the quality of life [1]. In novel smart cities applications, intelligence could be implemented in devices which allows to assign a level of autonomy. Autonomy opens important opportunities for collaboration and cooperation among these devices, such as cluster-wise cooperative automated trucks [2] and collaborative smart drones [3]. Using novel processing technologies, traffic components could collaborate to automatically detect traffic accidents [4] and road hazards. The collaboration could be useful not only to decrease task completion time or to coordinate the tasks but also to improve group awareness and increase productivity and efficiency 3]. In our previous work [5], we modeled collaborative join task planning in multi-UAV application where the actions of the drones are guided by a firefly algorithm using a reward and cost ratio. Perceived and received information in agent technology could be modeled as beliefs. Smart cities applications include different aspect of uncertainty. Co-existing agents could receive and perceive conflicting and uncertain information. With the scarcity of possible available resources, decision making cannot only be guided by the reward–cost ratio but should also take in consideration resource value and uncertainty of information. A variety of uncertainty management approaches have been proposed in literature. Different beliefs fusion operators could be used to fuse received and perceived beliefs. Whereas no single belief fusion operator is suitable in every situation [6]. The suitable belief operator depends on the situation to model, the conflict between uncertain beliefs, and the reliability attributed to source of information. 

In this research work, we aim to propose a belief management approach to model group beliefs under uncertainty in a drone collaboration for Intelligent Transportation System (ITS) application. We base this model on a probabilistic belief function to represent uncertainty. We propose a structure of individual agent belief base to model group and individual beliefs with different levels of uncertainty. To fuse perceived and received information, we use different belief fusion operators. The selection of the fusion operator is based on a Hierarchical Analysis Process (HAP). Moreover, to provide an in-demand belief-sharing service with a different level of uncertainty, we decompose agent belief bases in three repositories. In the application scenario, we aim to allow traffic agents (vehicles, drones, etc.) to select the most suitable fusion operators and combine received and perceived information with uncertainty, manage beliefs in the belief base, and provide group beliefs as services. In the following section, we present a short literature review of the main existing uncertainty management methods, belief fusions, and their combination with multi-criteria decision making (MCDM) methods. In the second section, we present the proposed agent dynamic belief management model. In the third and last section, we suggest an application of the proposed model in smart cities and the ITS domain.

## 2. Related Work

### 2.1. Uncertainty Management

The performance of smart city applications is commonly refrained by the insufficient management of uncertainty aspects. These aspects may concern, for example, data quality, communication reliability, and service availability. They are basically caused by noisy perceptions and/or observations, partial knowledge of the world, dynamic environments, etc. In order to manage this uncertainty, recent simulators are implementing new mechanisms to handle any vagueness caused by sensing devices, ongoing events, or networking infrastructures [7]. In spite of some relevant progress, a lot of research and development efforts are still needed.

Uncertainty can be classified into three main types [8]: perception-related uncertainty, data-related uncertainty, and epistemic uncertainty. Perception-related uncertainty is caused by the imprecision of sensing devices. Data-related uncertainty is caused by the limitations of data-driven components, which extract knowledge from information sources. Epistemic uncertainty concerns the presentation of pertaining information to be exchanged and processed by any collaborating entities. Uncertainty was also classified into aleatory and epistemic in [6,9]. Aleatory uncertainty is a type of statistical uncertainty that expresses that we do not know the outcome of an experiment. Instead, we only know the long-term relative frequency of outcomes. Epistemic uncertainty (also called systematic uncertainty) reflects that the outcome of an experiment could be known, without any evidence of its exactitude [6]. It can particularly occur when several potentially heterogeneous systems are required to collaborate. In artificial intelligence, the uncertainty has been categorized into uncertainty in prior knowledge, uncertainty in actions, and uncertainty in perception [10]. Uncertainty in prior knowledge is about facts and inference rules. Uncertainty in action is related to the pre-conditions of the actions. Uncertainty in perception is related to sensor data. In the specific context of collaborative systems, the management of uncertainty is of paramount important for an effective decision-making process. 

Several challenges are linked to uncertainty management. These issues particularly include its representation, identification, combination, and propagation. To represent uncertainty different models could be used. Probabilistic distribution has been used to deal with uncertainty problems. However, this will be costly in the case of continuous variables and hard to compute in cases of conditional distributions. Probabilistic graphical models have also been used to solve uncertainty [11]. Graphical models provide a powerful and flexible way to model relationships between variables and have been applied with great success [12]. Graphical models have also been suggested to model beliefs. An overview of the most common belief graphical models is presented in [11]. Automated uncertainty handling could be allowed via the qualification of uncertainty using probabilistic, fuzzy approaches, or rough sets [13]. Fuzzy sets are based on calibrating concepts and linguistic ambiguities. Probabilistic models use different probability distribution functions and statistics for modeling and could be used in aleatory and epistemic uncertainty [9]. Rough sets are mathematical tools to handle uncertainty and incomplete information. Sharing context information could also be used to jointly mitigate uncertainties [8]. Dempster–Shafer theory (DST) is widely applied to uncertainty modelling and knowledge reasoning because of its advantages in dealing with uncertain information [14]. These uncertainty models have a greater expressive power than probabilistic ones; however, they are more complex and often have higher computational cost [15]. DST (Evidence Theory or Belief Function) is effective in uncertain information modeling and processing, and it has been widely used in many fields. Figure 1 shows the relationship between main uncertainty theories. Probability theory and fuzzy theory are the most common theories to model uncertainty [9]. Probability theory can deal with both aleatory and epistemic uncertainty and is the most widely used method in almost every field. The review in [9] examines further existing uncertainty theories. A variety of approaches have been developed based on the classical Probability theory such as Monte Carlo method, Bayesian method, Subjective Logic, and extended models of DST. The DST belief model is highly expressive and extends the notion of probability while Monte Carlo is considered not suitable for projecting epistemic uncertainty through a complex model [16]. The Bayesian approach incorporates the uncertainty by model averaging. Bayesian model averaging (BMA) addresses the problem of model selection not by selecting a final model, but rather by averaging over a space of possible models that could have generated the data [17]. The Transferable Belief Model (BTM) concerns the same problem as the one considered by the Bayes model, except it does not rely on probabilistic quantification but on belief functions. Subjective Logic is an uncertain probabilistic logic that was initially introduced by Audun Josang to address formal representations of trust. In Subjective Logic (SL), the subjective opinion model extends the traditional belief function model of belief theory in the sense that opinions take base rates into account, whereas belief functions ignore base rates. SL could be used for probabilistic reasoning under uncertainty and model both probability and uncertainty. There are different models of uncertainty resulting from applications of probability theory such as Bayesian Belief Networks, Markov Decision Process Graph, Goal function roadmaps, etc. The Bayesian Belief network helps to model conditional dependencies using a directed acyclic graph. Bayesian Networks (BNs) have gained popularity in environmental risk assessment because they can combine variables from different models and integrate data and expert judgment and they are able to model conditional probabilities. Aleatory BNs are the most relevant for environmental risk assessment, but they are not sufficient to treat epistemic uncertainty alone because they do not explicitly express the parameter uncertainty [18]. In [18], the authors recommend embedding an aleatory BN into a model with parameter uncertainty for risk assessment. In addition to evidence theory, entropy has been used as a typical method for uncertainty measure and management [19,20,21,22]. In evidence theory, the belief entropy or uncertainty measure of mass function or basic probability assignment (BPA) is used to address the information volume of evidence [19]. A variety of entropies exist, such as Shannon entropy and Deng entropy, which is a generalization of Shannon entropy in the Dempster–Shafer framework [22]. The Belief Function Theory has shown its ability to model uncertain knowledge. It allows knowledge combination obtained through various sources of information. However, some requirements, such as exclusiveness hypothesis and completeness constraints, limit the development and application of DST theory to a large extend [14]. To overcome the shortcomings and enhance its capability of representing the uncertainty, a new mathematical tool to represent unreliable information, named D numbers, was proposed by Deng in 2012, which mirrors the framework of D-S theory. Compared with the classical DST, D numbers abandon the exclusiveness hypothesis of the frame of discernment (FOD) and mass function’s completeness constraint in DST, which make D numbers more reasonable and capable of modeling the ambiguous, imprecise, and vague information [14]. The D numbers, as a reliable and effective expression of uncertain information and has a good performance to handle uncertainties such as imprecision and incompleteness [23]. Other theories have also been proposed in literature as extension of on existing uncertainty theories [20,21,22].

For our work, we need to model epistemic uncertainty in the context of traffic information in smart cities. To handle uncertainty, we will adopt uncertainty mass and belief mass distributions of subjective opinions, which are based on Probabilistic theory. Subjective opinions could be mapped to the belief mass and belief mass distribution of DST since we do not focus on conditional probabilities. In the aim of simplicity, we assume in this paper that if an event has occurred, collaborating partners will inform entities in the same collaborative team. In addition, we do not address communication uncertainty, which means we do not represent in this work conditional probability. 

In SL, opinions on a binomial variable are called binomial opinions. A binomial opinion about the truth/presence of value x is the ordered quadruplet with additivity requirement as shown in Equation (1). Binomial opinions can be mapped to Beta Probability density function (PDF) which allow us to apply subjective-logic operators (SL operators) to Beta PDFs and allow to apply statistical operations of Beta PDFs to opinions [6]:ωx = (bx, dx, ux, ax), where bx + dx + ux = 1(1)

bx: belief mass in support of x being TRUE (i.e., X = x);dx: disbelief mass in support of x being FALSE (i.e., X = x);ux: uncertainty mass representing the vacuity of evidence;ax: base rate, i.e., prior probability of x without any evidence.

### 2.2. Beliefs Fusion

The terms belief and knowledge have been used in several research works as interchangeable terms. However, in [24], the authors emphasize that knowledge takes the format of sentences or formulas, while belief is knowledge with a degree of belief. This degree of belief is usually quantified by a probability measure, as it is the case in the Bayesian approach. Beliefs could be modeled as a belief function. The Belief Function Theory was initiated by the work of Dempster on the upper and lower probabilities. The development of the theory formalism is due to Shafer, which has shown the benefits of belief functions theory to model uncertain knowledge. In addition, it allows knowledge combination obtained through various sources. In an event-based approach, belief is defined in term of events with degree of uncertainty [25].

Referring to [6], belief fusion means precisely merging multiple opinions in order to produce a single opinion that is more correct (according to some criteria) than each opinion in isolation [6]. In belief fusion, the source of the opinions could be agent or sensor producing data that could be translated to opinion. The most common representations of belief states use sentences or propositions [26]. In the most common case, beliefs are represented by sentences or predicates. To tackle the problem of belief changes or belief revision, there are two general strategies to follow [26]: (1) Direct: inserting or deleting a unit of beliefs database without bothering about any integrity constraint; (2) Logic-constrained belief revision, which takes the integrity constraints as constraints for every process of belief change.

Belief function gives analysts the ability to assign belief mass to elements in the powerset of the state space [6] and explicitly express ignorance. Combination rules or fusion operators are usually addressed in uncertainty theories and differ for each uncertainty theory. The DST includes combination rules. However, the DST combination rule often obtains results contrary to common sense when it fuses highly conflicting evidence [27]. Some scholars have proposed to modify DST combination rule and others proposed to modify evidence sources. In addition, a variety of modified combination rules [28] and fusion approaches [20] were suggested based on distance and similarity. As an alternative, Deng proposed the basic framework of generalized evidence theory (GET) and pointed out that the traditional DST was unable to deal with information fusion problems when the frame of discernment (FOD) was incomplete [28]. Deng entropy has been widely used in many applications such as risk analysis. In GET, uncertainty is modeled by the concept of generalized basic probability assignment (GBPA). A generalized combination rule (GCR) is provided for the combination of GBPAs with a generalized conflict model (GCM) to measure conflict pieces of evidence. In D numbers, the combination of D numbers does not maintain the associative property. A method to efficiently combine multiple D numbers was proposed [24] using Multiple-Criteria Decision Making (MCDM), and others suggested modified combination rules for D numbers [14]. Many studies of evidence distance (e.g., Jousselme evidence distance) and evidence similarity (e.g., pignistic probability distance) emerged, aiming to evaluate the effectiveness of evidence combination algorithms. In subjective logic, the main belief fusion operators are:Belief Constraint Fusion: suitable for opinions with totally conflicting opinions or total uncertainty. This operator is suitable if the agent needs to believe only the common beliefs. If the agent has no common beliefs, the agent will not believe in both opinions.Average Belief Fusion: suitable when observations arrive at same time with different uncertainties.Cumulative Belief Fusion: suitable if observations arrive at different times with the same or different uncertainties.Weighted Belief Fusion: suitable for fusing agent opinions in situations where the source agent confidence (cx = 1 − ux) should determine the opinion weight in the fusion process. This means that the opinion of the agent with less uncertainty must guide the fusion process (due to the importance of the opinion of such agent: e.g., distance to the event).Consensus and Compromise Belief Fusion: suitable for transforming conflicting beliefs into compromising vague beliefs. The agent will compromise to adopt one of the beliefs with a reduced certainty.

These beliefs operators provide a flexible framework to model situations where there is uncertainty [6]. None of these operators will be appropriate for every decision-making task. However, it is promising to allow an agent to select the most appropriate operator for a particular decision task [6]. An approach to select a suitable fusion operator has been suggested in [29], where the authors propose a two-step fusion process based on the risk to guide decision based on imperfect information in natural hazards. In this research work, the authors propose a methodology to help decision making by combing the Analytic Hierarchy Process (AHP) with information fusion using Belief Function Theory. A methodology named ER-MCDA (Evidence-Reasoning–Multi-Criteria Decision Analysis) mixing MCDA and information fusion has been proposed to help to handle imperfect information and select the correct belief. The principle of the ER−MCDA methodology is to use the multi-criteria decision analysis framework to analyze the decision problem and to identify the criteria involved in the decision.

### 2.3. Collaborative Belief Management Frameworks

In collaborative teams, awareness about the situation is essential, MAS is one of the best approaches to model collaborative entities in different application domains. Awareness refers in its minimal form to an agent’s beliefs about itself (intra-personal), about other agents (inter-personal) and about the environment. Many logical models of an agent as an individual, autonomous entity have been proposed and successfully implemented. One of these well-known and successful models is the BDI agent model. However, when modeling collaborative teams, the agent individual model does not suffice for teamwork. When a team is supposed to work together in a planned and coherent way, it needs to present a collective attitude over and above individual ones [30]. One of the interesting problems in multi-agent research is the fusion of approximate information from heterogeneous agents with different abilities of perception and with uncertainty to avoid false beliefs [30]. Some existing works [30,31,32] proposed number of formal and logical models of teamwork addressing issues such as mutual intentions, shared plans, cooperative problem solving and mutual beliefs [30], but a few of these models addressed the uncertainty at the same time. For our research work, we particularly focus on mutual beliefs. Some of these research works focused on shared mental states and group beliefs. There are two aspects of shared beliefs. In the first aspect, agents may all believe a fact without knowing whether or not others believe the fact. In the second aspect, the agents believe a fact and know that other agents also believe the fact. In [30], the proposed belief model has three levels: individual beliefs, general beliefs, and common beliefs. The general belief means that every agent in the group believes in ϕ. Common belief means everyone in the group believes ϕ, everyone in the group believes that everyone in the group believes ϕ, etc. The Common belief is stronger than general belief and requires agents to have an awareness of the beliefs of other members. However, common beliefs and general beliefs need not be true. Other terms that are used for shared beliefs are mutual beliefs, mutual knowledge, and shared knowledge [33]. Mutual belief is expected to be an important diagnostic for defining communities of interest [31]. In addition, sharing beliefs was used in team cognition, which describes the mental states that enable team members to anticipate and to coordinate [34] and also refers to the representation and distribution of team relevant knowledge within teams [35]. Existence of such team level knowledge enables members to anticipate and execute actions. Team cognition impacts at least the team processes and team performance. In distributed team cognition, it is important to have a trustworthy communication channel for a cooperative awareness because unreliable communication could cause doubt in message receptions. Another model for collaborative awareness of an agent was proposed in [32,35] where the collaborative agents share a team mental model where the mutual belief is defined via an ‘everyone believes’ operator, which states that every agent in group believe in ϕ. The authors suggested a new architecture for agents with Shared Mental Model (SMM) components and key processes. SMM refers to a common understanding by the team members regarding tasks, the team, and temporal aspects of their work [35]. One of the multiple functions of SMM is to allow the team members to interpret information in a similar manner and share expectations concerning future events. The architecture proposed includes the shared mental model and the belief base with the SMM Update Handler. The handler serves multiple functions, such as the use of communicated information to update the SMM and to generate new beliefs. In the proposed architecture, the SMM interacts with the Goal Reasoner and Planner to generate the next action in the plans. However, the proposed model does not include uncertainty of perceived and received information. In addition, the flow of beliefs from the belief base to the SMM is unidirectional, which means once the belief is in the SMM, it will not revert to the belief base. The SMM has also been used for coordinating agent and human cooperation [36]. However, in this work, the SMM is particularly used to interact efficiently with other team members and to track progress in terms of goals, subgoals, planned and achieved states, and other team-related factors [36]. In addition, in [37], the SMM was used as a distributed knowledge base to coordinate activity and make team-oriented decisions, but for robots and humans. For our case, the coordination between agents is performed via the Mission Leader Agent by communicating the Capability Map [5]. We particularly focus on the beliefs related to the reporting important events with uncertainty which affect planning and decision making. Shared Beliefs under conditions of uncertainty has been suggested in [38]. However, the authors focus on the formal semantics model and integrate uncertainty by adding a probability estimation. The ability to reason about beliefs, goals and intentions of other agents and to predict those of opponents/partners is called Theory of Mind (ToM). The concept of shared beliefs has also been discussed in ToM with the consideration of uncertainty [38], in which shared belief was used as alternative of common knowledge.

In collaborative team network, message passing is one of the methods that has been exploited to model belief propagation in an uncertain environment [39,40]. A variety of belief propagation approaches has been suggested in the literature, such as Loopy Belief Propagation (LBP), Weighted Loopy Belief Propagation (WLBP), Generalized Belief Propagation (GBP), Nonparametric Belief Propagation, Weighted Loopy Belief Propagation, belief propagation-based dead-reckoning (BPDR) [40], etc. Belief propagation was exploited for variety of applications such as distributed inference [41], cooperative sensing [39], cooperative localization [40,42], collaborative navigation [43], multi-Drone monitoring [44], etc. The performance of these approaches depends on the topology of the graph, as well as the characteristic of the application domain (large-scale applications). Belief propagation has also been used recently to solve uncertainty in cooperative self-organized drone swarms [45].

### 2.4. Multi-Criteria Decision Analysis and Uncertainty

The decision-making process is a core component in modern IoT applications. A variety of machine learning algorithms and framework can be used to derive decisions and improve a variety of operational aspects, such as planning [46], scheduling [47], resource allocation [48], etc. Multi-criteria decision making (MCDM) is one of the most well-known branches of decision making. MCDM is divided into multi-objective decision making (MODM) and multi-attribute decision making (MADM). However, very often the terms MADM and MCDM are used to mean the same class of models [49]. MODM studies decision problems in which the decision space is continuous. MCDM/MADM concentrates on problems with discrete decision spaces. In these problems the set of decision alternatives has been predetermined. The weighted sum model (WSM), the analytic hierarchy process (AHP), the revised AHP, and the weighted product model (WPM) are examples of MCDM. One of the most important aspects of the AHP method is the organization of the problem in a systematic way, such as by goal, criteria, and alternatives, that provides a structured simple solution to decision making problems [50]. In AHP, it is important to take into consideration uncertainty in multicriteria decision making. At the same time, DST has received the considerable attention of different researchers whom explored its potential in various domains. In 2000 Beynon et al. develop a methodology in the field of decision making that incorporates DST with the AHP to solve complex problems involving multiple criteria. In [51], the authors propose to improve classical multi-criteria decision making by integrating uncertainty theories such as fuzzy sets, possibility, and belief function theories using a three-fusion process. In this work the proposed approach has been applied to natural disaster taking in consideration variant criteria like the sensitivity of the avalanche prone area. A variety of MCDM models have been combined with uncertainty theories. In [52], the authors suggested to develop an Elimination and Choice Translating Reality (ELECTRE)-based MCDM method where the evaluation of information is expressed and handled by a DST. In [26], the authors propose an MCDM method based on D Numbers and belief entropy. AHP, essentially, is the process of assigning different weights to different options and summing them up. In this work, we calculate the weights based on meta-data of received and perceived information and the quantification of subjective beliefs.

In MAS, some recent research works integrate evidence theory and its fusion in MAS [30]. The AHP and DST has been combined for fuzzy multiple criteria group decision-making in order to improve the accuracy of selection of new products in the context of product development. These hybrid approaches combining MCDM and uncertainty theories have been exploited in transportation [50] and logistics [53].

## 3. Hierarchical Analysis Process for Intelligent Collaborative Belief Management

In this work, we propose to model the beliefs of the collaborative group in a decentralized manner. In the environment, different collaborative teams could be created on demand based on the need as explained in our previous work [5]. Each team has a team leader agent, which initiates the collaboration process, and member agents, which accept or refuse to join the collaboration. Once an agent accepts to join a group, he became a member. The team members will execute sub-tasks of the global mission. The leader agent will serve as a group belief repository for the team, which allows the team members to subscribe to receive reported information. The beliefs contained in the team leader are represented in four repositories:Temporary Beliefs: This beliefs repository contains the beliefs received from the team members. The team members will report collected information or events to the team leader. Once the team leader receives this information, he will store this information in the Temporary Beliefs repository.Individual Beliefs: This beliefs repository contains the beliefs perceived by the agent.Promoted Beliefs: This beliefs repository contains the beliefs frequently reported by different team members.Shared Beliefs: This belief repository contains the beliefs in which all the team members belief on it. These beliefs could be considered as an alternative model to model common beliefs presented in [28].

Figure 2 represents the belief base of the team leader agent. The collaborative team members have the same model of belief management, except that they do not store temporary beliefs from the team members of the collaborative group to reduce communication. Based on their subscription to the geographical area of interest on a selected repository, the collaborative team members will receive new reported information in the specified beliefs repository. This model of belief management allows collaborative team members to benefit from being a part of the group and increase awareness. At the same time, it evaluates the uncertainty of the information and allows member agents to subscribe to specific repositories. The reported information could also be used by the collaborative agent to guide his decision-making process. The collaborative agent has a degree of freedom to choose between selfish or collaborative behaviour [5]. However, the members will not benefit from the group awareness if they decide to disband the team.

To allow all members of the collaborative team to benefit from this beliefs model, create a team awareness, and reduce the communication messages, we allow each team member to subscribe to a region of interest. Each agent interested in receiving the information reported about a region will subscribe to the region of interest in the leader agent. The leader agent will also include a module to handle the subscription of the agent members. The team members could subscribe to temporal, promoted and shared beliefs. All the team members which will execute a task in the region of interest will publish the reported information to the leader agent, which will store it in the temporary belief repository. Based on the certainty and the frequency of the reported information. The team leader can then decide to store the belief in the promoted or shared repository or keep it in the temporary beliefs. Figure 3 represents the architecture of collaborative team of agents.

To manage the uncertainty of perceived and received information, the agents will use subjective logic to create opinions about the perceived environment. The leader agent will receive reported information and use an HAP module to identify the operator that should be used to fuse the perceived and received opinions. The proposed module uses the following criteria to decide the operator that will be used:Time: The time difference between the perception of the first opinion and the time of the perception of the second opinion;Location: The location of the agent reporting the opinion. The location of the agent will be used to calculate the distance between the agent reporting the event and the team leader;Source: The agent sending the opinion. The source will be used to find the trust in the source. The evaluation of the trust is out of the scope of this work;Risk: The risk of the perceived and received information. The risk could depend on the nature of the region where the event is perceived or the type of the perceived event or hazard;Uncertainty: The uncertainty of the perceived and received opinions;Conflict: The conflict between the perceived and received opinions.

The HAP module will take the five SL operators as alternatives. We define the weight of each criterion before the execution of the drone’s mission. The weights could be defined from the literature, from domain experts or derived from a training phase. These weights could be refined after the execution of the mission. During the mission, for each received information, the leader drone agent will evaluate the criteria (time, location, source, risk, uncertainty, and conflict) using the received and perceived information to decide the operator or strategy that should be used to combine the opinions. The selected operator will be used to fuse two opinions and produce a new opinion, as presented in Section 2.2. The selection of the operator will affect the uncertainty of the resulting opinion. Since the leader opinion will be shared with the team member’s drone, the selected operator will also affect the state of the belief bases of the drone members and their actions.

## 4. Implementation in the Context of Intelligent Transportation Systems

We are focusing in this paper on the collaborative operations of an Internet of Drones in the context of ITS.

### 4.1. Formulation

To model uncertainty, we use Equation (1) from subjective logic. We model the environment as a grid of cells. Several events could be randomly created and diffused in the environment. The events could also disappear after a certain time. In the environment, we model drones as Belief–Desire–Intension (BDI) agents. Each drone will have parametrized characteristics, such as the capacity of the battery, the range of the field of view, and the charging time. The agent drone will be able to perceive only the cell in his field of view. Once an event is perceived, the agent will calculate the opinion based on the base rate, the belief mass, the disbelief mass, and the uncertainty and send the opinion to the agent leader.

In the belief module a hierarchical structure model will be built to decide the best fusion operator, as shown in Figure 4. First, a Pair-Wise Comparison Matrix Criteria weights (Matrix 1) is represented to identify the importance of each criterion compared to the others. In our case, we choose to construct a judgement matrix with the (1)–(9) scale method:(2)$$V=\{V_{time,\;}V_{location\;},V_{risk},\;V_{trust},\;V_{uncerainty},V_{conflict}\}\;A=\left[\begin{array}{ccc} a_{1,1} & \cdots & a_{1,j}\\ \vdots & \ddots & \vdots \\ a_{i,1} & \cdots & a_{i,j} \end{array}\right]\;(\mathrm{Matrix}\;1)$$

To evaluate the consistency index of the proposed matrix, we process the following: steps.

We calculate the judgement matrix normalized by column:


(3)
$$b_{i,j}=a_{ij}/\sum a_{ij}$$


2.The normalized matrix is summed by row:


(4)
$$c_{i=}\sum_{j=1}^{n}bij$$


3.$c_{i}$ is normalized and weights are obtained


(5)
$$w_{i}^{2}=c_{i}/\sum c_{i}$$


4.Find the maximum eigenvalue corresponding to weight;


(6)
$$w^{2}:\;\lambda_{\max}=\frac{1}{n}\sum_{i}\left(\frac{{\left(AW^{\left(2\right)}\right)}_{i}}{w_{i}^{\left(2\right)}}\right)$$


5.Consistency testing: We calculate the degree of inconsistency or Consistency Index (CI) of the matrix A to make sure that the rankings given by different decision makers and used as inputs to the AHP application are consistent:


(7)
$$\mathrm{CI}=\frac{\lambda_{\max-n}}{n-1}$$


6.We finally calculate the Consistency Ratio (CR). The ratio of Consistency Index (CI) and the Random Consistency Index (RCI). The CI measure the degree of inconsistency. The larger the inconsistency between comparisons, the larger the consistency index. The comparisons should have a much lower consistency index than what would be produced by random entries. The RCI is the mean CI for random entries. The RCI is defined for different sizes of the matrices [49]. For our case the size of the matrix is six, which means the RCI = 1.24:

CR = CI/RCI(8)

Saaty [54] states that an acceptable consistency ratio should be less than 0.1, yet a ratio of less than 0.2 is considered acceptable. To calculate the benefits of each strategy, we define the utility values for each strategy. The value of each utility change for each strategy. For example, the belief constraint fusion operator is suitable when opinions are totally conflicting or totally uncertain. So, the conflict and uncertainty are the most beneficial criteria. For Average belief fusion, time criterion is the most important than uncertainty. Following Table 1, for cumulative belief fusion, the value *t_cum* is greater than *t_avg*, *t_const*, *t_wg*, and *t_comp*.

The total of the utility for each operator (example: Average Belief Fusion) is calculated using the Equation (9) and the benefit of each option is calculated using the Equation (10):(9)$$u_{avg}=\frac{1}{\left(t_{avg}+d_{avg}+c_{avg}+u_{avg}\right)}+\left(r_{avg}+tr_{avg}\right)$$
(10)$$B_{op}=\left[\begin{array}{c} w_{1}^{2}\\ \ldots \\ w_{n}^{2} \end{array}\right]\times \left[\begin{array}{c} u_{1}\\ \ldots \\ u_{n} \end{array}\right]$$

For each operator, the benefits will be evaluated and the operator with the maximum benefits will be selected from the candidates. After the selection of the operator, the leader agent will fuse the two opinions and produce a new opinion. This opinion will be stored in the temporal belief. Based on the received information, this belief could afterward be moved to a promoted belief or the shared belief repository.

### 4.2. Simulations and Results

To show the utility of the proposed belief management module, we here present an example of the application in ITSs where the environment contains vehicles, road infrastructure, roadside units, drones, etc. For simplicity, the environment will be divided in a grid-based decomposition. Each drone is assigned to a region as presented in Figure 5. In each of the region one or many roadside units will be placed. The vehicles will report traffic events to the roadside unit. The roadside units will report the information to the closest drone, which reports the event to the team leader drone. The leader drone will use HAP to combine the received information and the perceived information and generate new belief. Then, it will use a decision module to decide the repository to store the belief. Once the drone identifies many traffic events that should be monitored which exceed his capabilities, the drone will request collaboration from drones within his communication range and share sub-tasks with the team members. Once the drones accept the collaboration, they will be able to select the best subtasks according to their states and capabilities. The drone team members will leave their regions and move to the region of the team leader to maintain adherence to the collaboration commitment.

The team leader drone will allow the drones which are members of the group to subscribe to the three repositories and share the stored information with interested members of the group to update the collaborative plan and take in consideration the new traffic events in the team member regions.

To simulate the described application, we used Agent-Based Modeling (ABM) to model the autonomy and the proactiveness of the drone and prototype the collaboration. We used BDI agents with GAMA simulator. We modeled the environment as a grid cell, and we defined two options for the distribution of the events. The first option is to generate a random number of event sources, as shown in the Figure 6b, or a single event with a random distribution (Figure 6a). We set three drones in the simulation, one leader drone and two drone members, and one charging station (yellow cube). The team formation of the collaborative network is out of the scope of this paper. In [5], we proposed an approach to form the team network. For each drone, we define a set of characteristics such as battery capacity, range of the field of the view, speed, etc. Each drone can perceive the events in his field of view. Once an event is perceived, the drone will calculate his opinions, and send it as a FIPA message to the leader. The leader will use the HAP module to select the operator that should be used to combine the different received and perceived beliefs.

We represented the view of the state of the world from the opinions of each agent (using his individual belief). The member drones will initially only be able to identify the perceived cells. Combining the received and perceived opinions, the drone leader will revise his temporal belief repository. The leader drone will publish the combined opinions to the team members. To show the difference between each fusion operator and proposed belief management module, we represent the state of the environment (Figure 7a) and the state of the individual belief of each agent (Figure 7c–e) and the temporal belief of the leader agent (Figure 7b). The intensity of the colour in the cells represents the value of uncertainty (higher intensity indicate a higher value of uncertainty). 

In the following, we track the changes of the mean uncertainty in the temporal belief of the drone leader based on the opinions received and the meta-data of the opinions (reception-time and location and the source of the information and risk) using the three operators of subjective logic (Cumulative fusion, Averaging fusion, and Weighted fusion). The cumulative fusion operator assumes that the amount of independent evidence increases and the uncertainty decreases by including more and more sources. As depicted in Figure 8, at the beginning of the simulation, the team leader operates alone without the collaboration of the member’s drone. After the diffusion of the event and the participation of the other drone. The mean value in the temporary belief base of the leader drone decreases by involving more sources. The value continues to decrease even with an important time difference between the last perceived/received state of the cell and the received opinion about the same cell.

The average belief fusion assumes that including more sources does not mean that more evidence is supporting the conclusion. Assume that agents A and B observe the same outcomes of the same process over the same time period, so their opinions are necessarily dependent. However, their perceptions might be different (e.g., because their cognitive capabilities are different). The average operator is suitable when no prior knowledge is given for the reliability of each source. So, the opinions of the sources are considered equally reliable. As depicted in Figure 9, the mean uncertainty in the temporal beliefs of the leader drone first decrease then increase when time difference increases.

The weighted belief operator is suitable when the opinions should not have the same importance for deriving the resulting opinion. For example, if the agent receives an event from another agent that he trusts, even if he has a conflict with his current belief, he will adopt the opinion of the second agent. In this case, trust is more important than conflict. Another example is the risk of the event. If an agent receives information about an event with high risk (example: fire in industrial facility), more weight will be assigned to this opinion, even it has a low uncertainty to mitigate undesirable consequences. The reception of an event with a high risk should guide the agent to choose a weighted belief fusion operator in which the new opinion will follow the agent with less uncertainty. In the simulation, we evaluated the risk based on the distribution of the event in the neighbor cells of each cell. The value of risk will be in the range [0, 1].

Using a cumulative fusion operator, the mean uncertainty increased due to the participation of other drone members in the collaborative awareness (Figure 8). The usage of this operator will be beneficial for events reported with small time interval. When the time interval increases, the use of the average fusion operator will be more beneficial (The uncertainty increases when the time interval increases (Figure 9). However, the average fusion operator assumes that opinions have the same importance, which is not the case if one of the reported events has an important risk value, whereas the weighted belief fusion operator considers the risk value. Thus, the uncertainty decreases when the risk increases (Figure 10). In the realized simulation, the AHP module allows the agent to alternate between operators based on the distance difference, time difference, risk difference, and trust difference. As shown in Figure 10, using the AHP module, the agent adopted the cumulative fusion operator at the beginning of the simulation. Afterward, the leader switched to the average fusion operator. Receiving event messages with a high-risk difference, the leader adopted the weighted belief fusion operator, which decreased the uncertainty in the temporal belief of the leader drone. When the risk reduced, the agent switched back to the average belief operator. 

Using the belief management module, we allow the team leader agent to select the best operators and receive the benefit of each operator in the correct situation. For example, as presented in Figure 11. At the beginning, the leader drone used a cumulative belief fusion operator then switched to an average belief fusion operator. When the risk value increased, the drone used the weighted belief fusion operator. In the simulation, the leader will select the belief operator (strategy) that maximize the benefit referring to the Equation (8), in which he will evaluate the time difference, distance, risk, and trust.

After the drone selects the strategy to adopt, the agent will combine the received belief with existing beliefs. If the belief did not exist before, the leader agent will adopt it and add it to his temporal belief. The temporal beliefs will be cleared each time interval. The new belief will move to one of the leader drone belief repositories. For the selection of the belief repository, we base it on the frequency of the reported information to decide the transition from one belief repository to the other. The transition from a belief repository to another will be modelled as more sophisticated decision-making module in a future works. We model beliefs repositories as repositories that will be used by the agent to store the combined belief; this repository will be revised after the insertion of the new beliefs to insure the consistency of the belief repository.

## 5. Conclusions

One of the important aspects of the collaboration is the management of group knowledge and awareness. The smart cities and ITS applications are integrating more and more data from a variety of sources or IoT devices with different uncertainty. Now IoT devices can perform more processing onboard, such as the identification of events, vehicles, hazards. These sensors and equipment have heterogonous capabilities and can generate information with different uncertainties. Combining the opinions of group members provides a collaborative awareness. In some situations, increasing the number of sources of the information leads to a lower uncertainty. However, this is not always correct, since events could be dependents. The use of average uncertainty gives an equal importance to each opinion, which is also invalid in some situations. Many approaches have been suggested to manage the uncertainties. Combining MCDM with evidence theory has also been proposed in some research works. However, most of these proposed approaches use a single fusion operator and do not focus on the selection of the operator but on the selection of the opinion. In our work, we proposed an AHP module to decide the suitable fusion operator to use as a strategy to combine uncertain opinions, and we apply this model in an autonomous collaborative drone network. The selection of a correct fusion operator will guide the drone to make better decisions about his actions. In addition, the afforded leader drone belief repositories will provide the agents in the group the opportunity to receive continuous updates about the region with different levels of reliability. The created repositories could also be provided as services to other drone networks. We aim to extend the proposed belief management module empowering the leader agent by a decision mechanism to decide the transition of the belief from a repository to another and to provide the collected data to other inter-group collaboration. In addition, we aim to improve the stability of the proposed model using theories such as OODA (Observation, Orientation, Decision, Action) ring [55] or Game-Theoretic Utility Tree, which are suitable for adversarial environments [56].

## Figures and Tables

**Figure 1 sensors-22-06146-f001:**
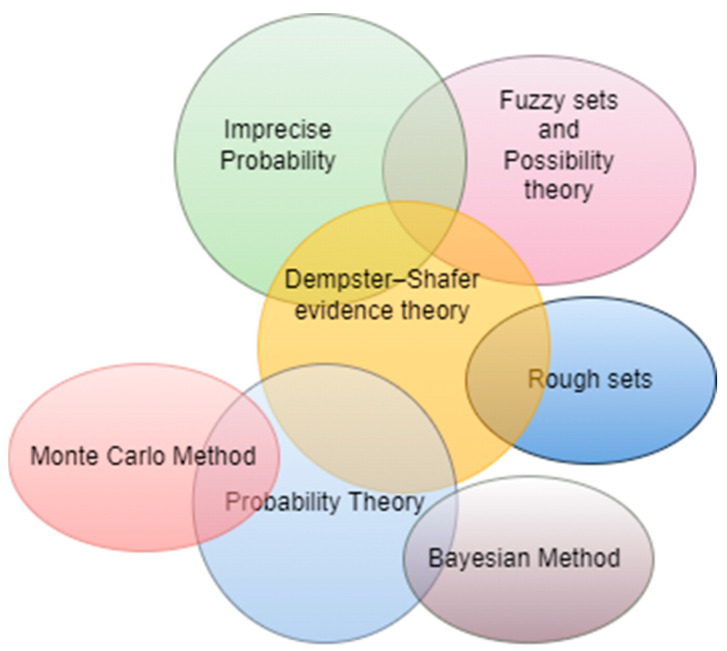
Relationship Between Uncertainty Theories.

**Figure 2 sensors-22-06146-f002:**
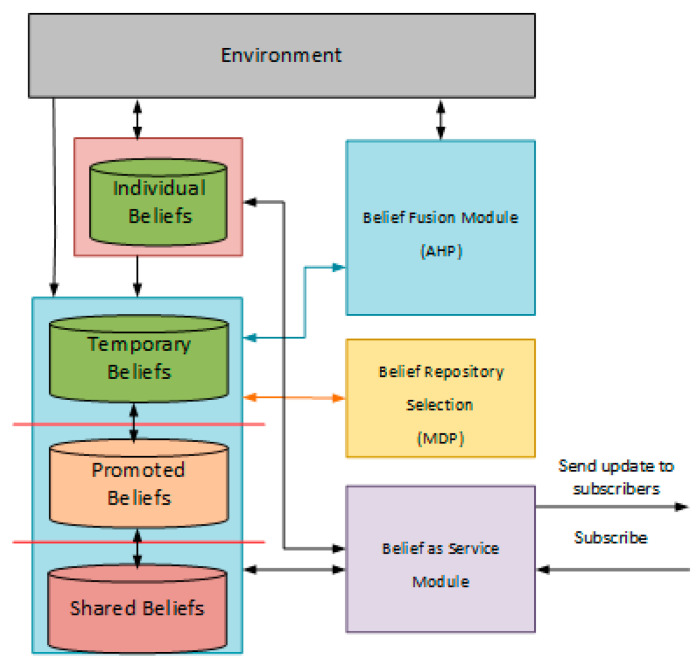
Representation of beliefs management module in the leader agent.

**Figure 3 sensors-22-06146-f003:**
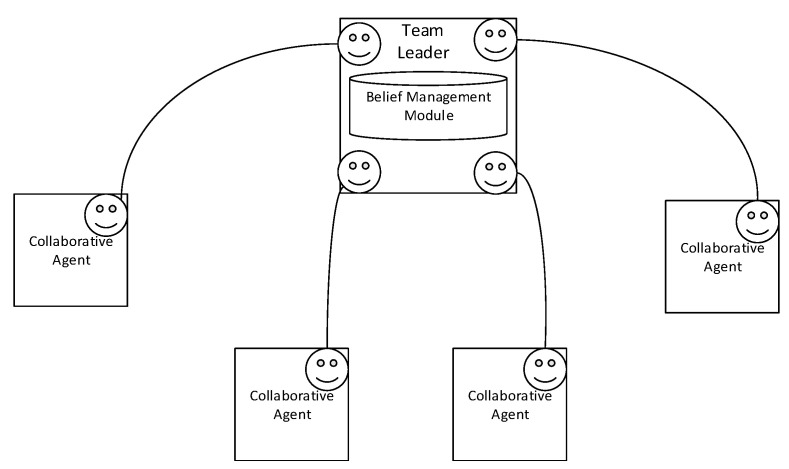
Decentralized management of team beliefs.

**Figure 4 sensors-22-06146-f004:**
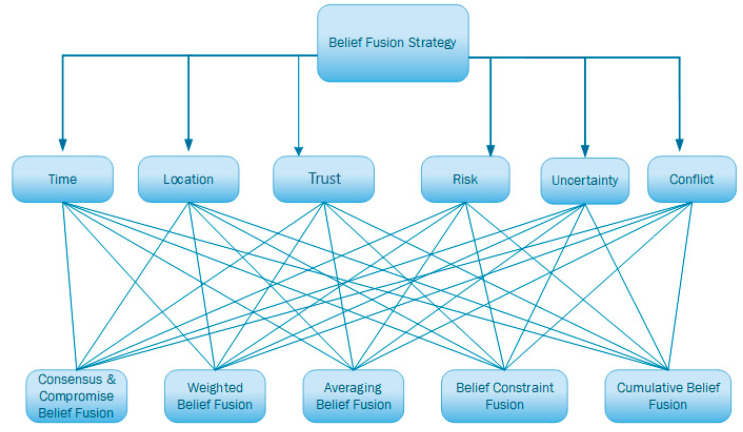
HAP for the selection of belief fusion strategy.

**Figure 5 sensors-22-06146-f005:**
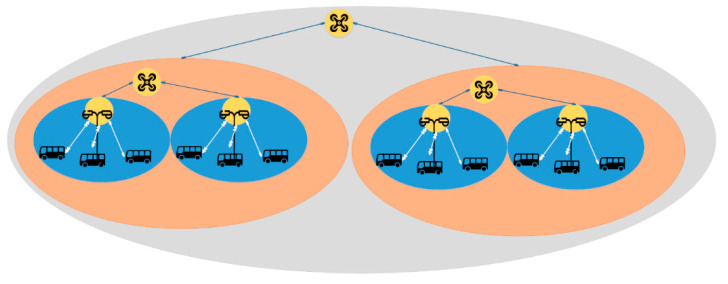
Simplified Architecture of belief management module application in ITS.

**Figure 6 sensors-22-06146-f006:**
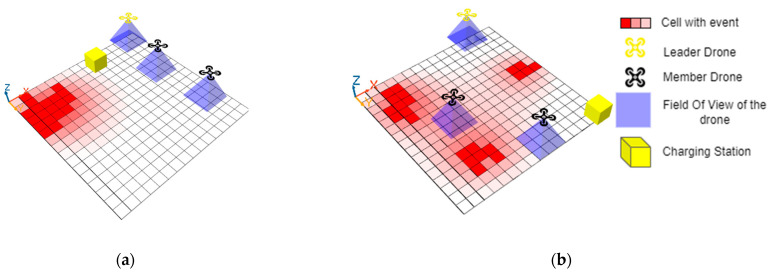
Simulation setting with (**a**) Single origin of event; (**b**) Multiple origin of event.

**Figure 7 sensors-22-06146-f007:**
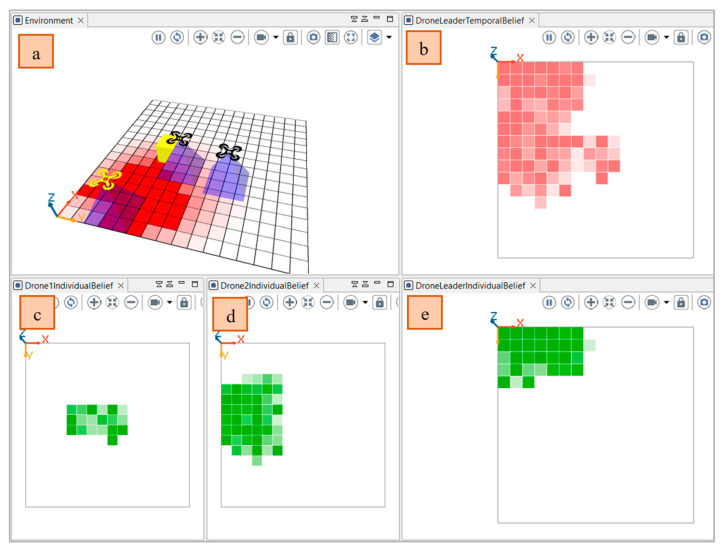
Simulated Environment and state of individual belief of each drone agent and the temporal belief of team leader agent: (**a**) Environment; (**c**,**d**) Individual belief of drone members; (**e**) Individual belief of leader drone; (**b**) Temporal belief of drone leader.

**Figure 8 sensors-22-06146-f008:**
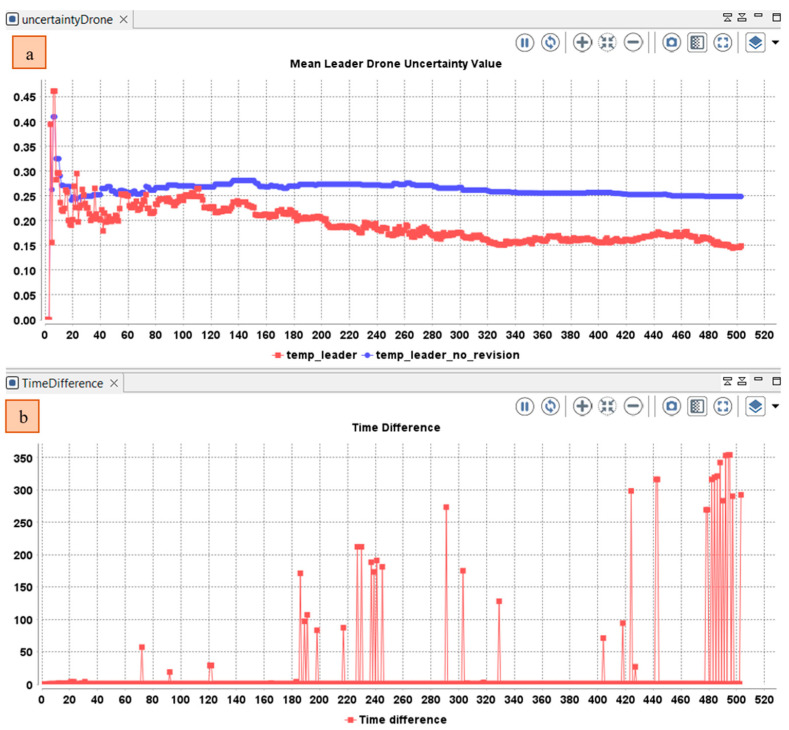
(**a**) Variation in the mean uncertainty over time in the temporal belief of the drone leader using only cumulative fusion operator; (**b**) Difference between the perceived and received opinions.

**Figure 9 sensors-22-06146-f009:**
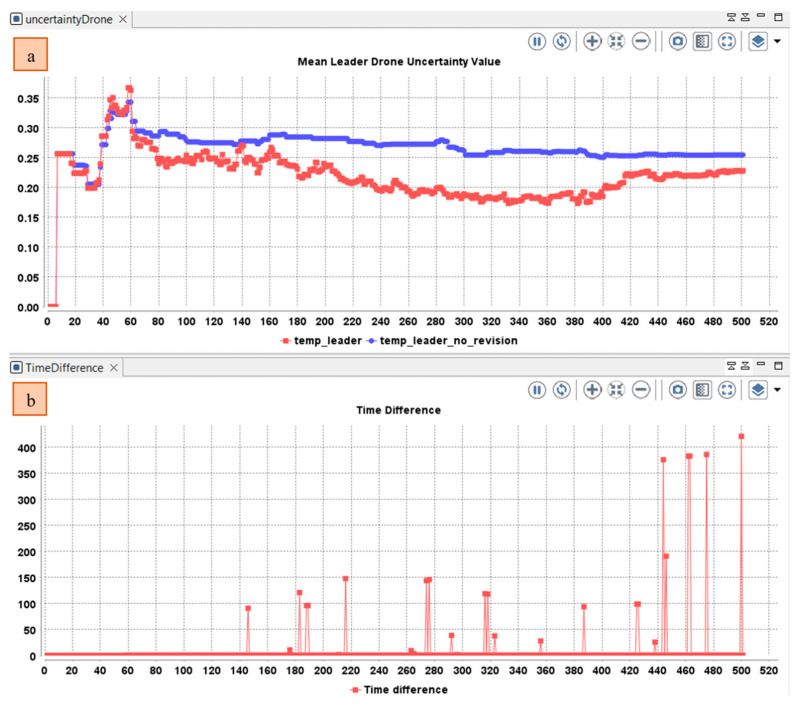
Variation in the (**a**) Mean uncertainty over time in the temporal belief of the drone leader using only average fusion operator; (**b**) Difference between the perceived and received opinions.

**Figure 10 sensors-22-06146-f010:**
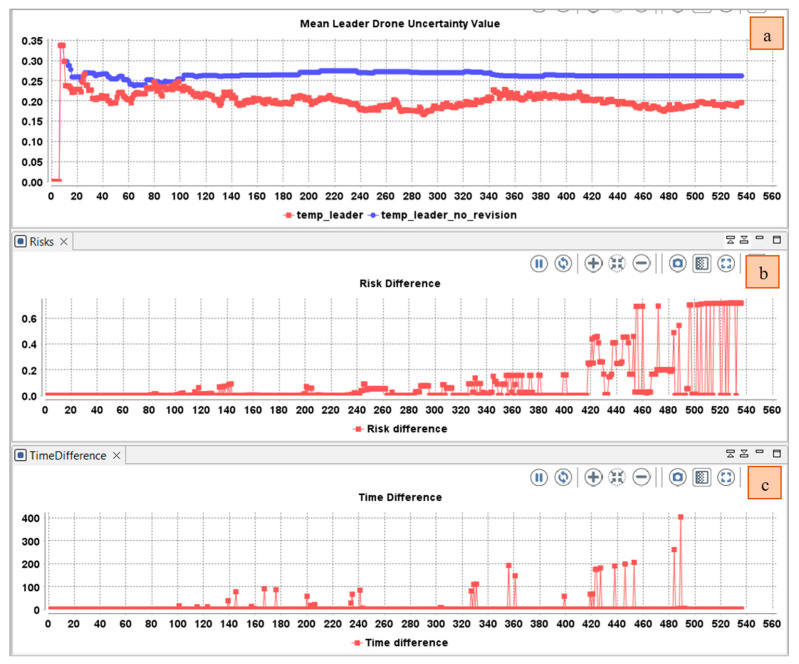
Variation in the: (**a**) Mean uncertainty over time in the temporal belief of the drone leader using only weighted fusion operator; (**b**) Time difference between the perceived and received opinions (**c**) Risk difference between the perceived and received opinions.

**Figure 11 sensors-22-06146-f011:**
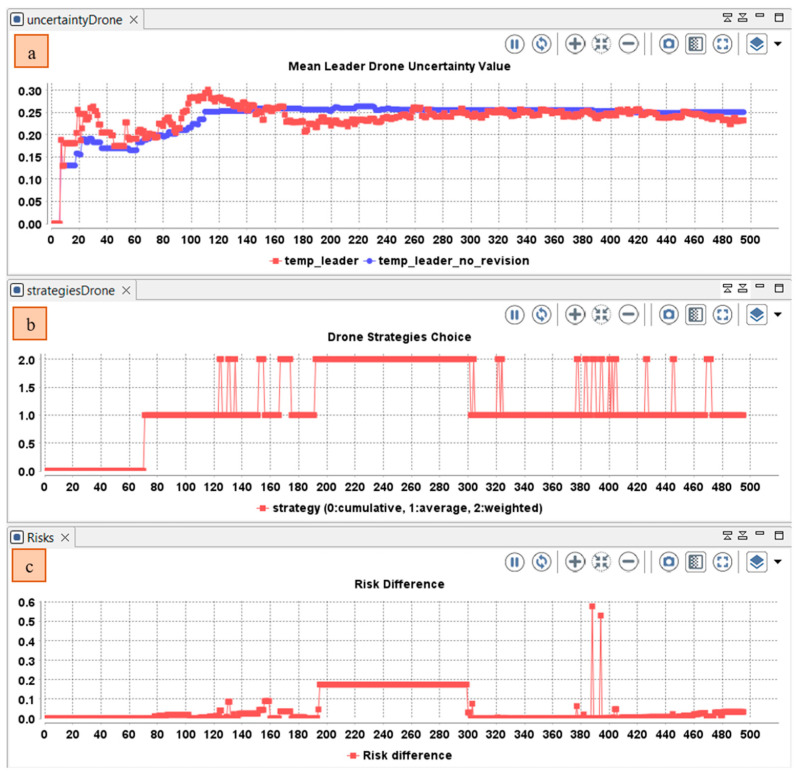
Variation in the (**a**) Mean uncertainty over time in the temporal belief of the drone leader using AHP module; (**b**) Variation of selected belief fusion operator in the leader drone (0: cumulative, 1: averaging, 2: weighted); (**c**) Risk difference between the perceived and received opinions.

**Table 1 sensors-22-06146-t001:** Assignment of criteria values for different alternative strategies.

Candidate	Time Difference	Distance to Event	Risk	Trust	Uncertainty	Conflict	Benefit
Average Belief Fusion	$t_{avg}$	$d_{avg}$	$r_{avg}$	$tr_{avg}$	$u_{avg}$	$c_{avg}$	$B_{avg}$
Belief Constraint Fusion	$t_{const}$	$d_{const}$	$r_{const}$	$tr_{const}$	$u_{const}$	$c_{const}$	$B_{const}$
Cumulative Belief Fusion	$t_{cum}$	$d_{cum}$	$r_{cum}$	$tr_{cum}$	$u_{cum}$	$c_{cum}$	$B_{cum}$
Weighted Belief Fusion	$t_{wg}$	$d_{wg}$	$r_{wg}$	$tr_{wg}$	$u_{wg}$	$c_{wg}$	$B_{wg}$
Consensus Compromise Belief Fusion	$t_{comp}$	$d_{comp}$	$r_{comp}$	$r_{comp}$	$u_{comp}$	$c_{comp}$	$B_{comp}$

## Data Availability

Not applicable.

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
