# Peer review of "Hierarchical Analysis Process for Belief Management in Internet of Drones"

_sensors, 2022, doi:10.3390/s22166146_

Round 1
Reviewer 1 Report
To model group beliefs under uncertainty, this paper proposes a Hierarchical Analysis Process to represent and combine opinions from different sources, and presents an example in ITS in the form of simulation to verify the model.
There are several issues that need to be improved in this paper:
1. In the evaluation for consistency index of Matrix 1,it is necessary to present the reason why RCI takes 1.24.
2. In section 4.1, how to determine the utility values of the given criteria for different operators is not explained.
3. The simulation results need further analysis. For example, the difference between Fig.11(a) and Fig.8(a)~Fig.10(a) should be explained in detail.
4. There are some errors in the paper.
i) The criteria in Fig.4 is “Source” while it is “Trust” in Table 1.
ii) In line 473, the table 3 is not found in the paper.
iii) In Table 1, the symbols in column “Conflict” and “Trust” should be checked carefully.
iv) In Line 491, there are two “received”. One of them should be “perceived”.
Author Response
We would like to thank the reviewer #1 for his/her valuable comments on the manuscript. To address the concerns raised, a Figure changed, an equation has been added and some modification has been made in the text. Changes to the text are underlined and colored in the track version.

Reviewer 2 Report
Recommendation
In this article, “Hierarchical Analysis Process for Belief Management in Internet of Drones." This paper presents a new belief management-based model for collaborative Internet of Drones. The topic is very important, but the manuscript is not clear. Also, the authors didn't try to improve the contribution of the system mode and methodology compared with the previous related works. The English language is acceptable and sufficient. However, I have some comments and suggestions for the authors. The authors should improve their manuscript as follows:
1- The authors must write the full meaning words of “ITS” in the beginning
2- How can enable the Collaborative agent to take the decision-making process in real-time.
3- How can produce a new opinion without using training based on the time difference between the perception of the first opinion and the time of the perception of the second opinion.

Author Response
We would like to thank the reviewer #2 for his/her valuable comments on the manuscript. To address the concerns raised, some modification has been made in the text. Changes to the text are underlined and colored in the track version.
Round 2
Reviewer 2 Report
Recommendation
In this article, “Hierarchical Analysis Process for Belief Management in Internet of Drones. The author improves the manuscript. But the manuscript is not clear. Also, the authors didn't try to improve the contribution of the system mode and methodology compared with the previous related works. The authors still haven't answered or checked comments no. (2), and (3). However, I have some comments and suggestions for the authors. The authors should improve their manuscript as follows:
1- How can enable the Collaborative agent to take the decision-making process in real-time.
2- How can produce a new opinion without using training based on the time difference between the perception of the first opinion and the real- time of the perception of the second opinion.
3- How can improves training stability in the internet of Drones based on your proposal to estimate the value of every action for certain states and the expectation of the action-value distribution.
Please see the citation references :
1- Jo TH, Ma JH, Cha SH. Elderly perception on the internet of things-based integrated smart-home system. Sensors. 2021 Feb 11;21(4):1284.
2- Salh A, Audah L, Kim KS, Alsamhi SH, Alhartomi MA, Abdullah Q, Almalki FA, Algethami H. Refiner GAN algorithmically enabled deep-RL for guaranteed traffic packets in real-time URLLC B5G communication systems. IEEE Access. 2022 Apr 25;10:50662-76.
3- Salh A, Audah L, Alhartomi MA, Kim KS, Alsamhi SH, Almalki FA, Abdullah Q, Saif A, Algethami H. Smart Packet Transmission Scheduling in Cognitive IoT Systems: DDQN Based Approach. IEEE Access. 2022 Apr 25;10:50023-36.

Author Response
We would like to thank the reviewer for his/her valuable comments on the manuscript. To address the concerns raised, some modification has been made in the text and some references have been added. Changes to the text are underlined and colored in the track version.
